# Sustainment of a patient flow intervention in an intensive care unit in a regional hospital in Australia: a mixed-method, 5-year follow-up study

Eva Ros ,[1,2] Axel Ros ,[2,3] Elizabeth E Austin ,[2] Lina De Geer,[1,4] Paul Lane,[5] Andrew Johnson,[5,6] Robyn Clay-Williams [2]

For numbered affiliations see end of article.

**Correspondence to**
Ms Eva Ros;
evaro626@student.liu.se

## ABSTRACT

**Objective** In 2014, an escalation plan and morning handover meetings were implemented in an intensive care unit (ICU) to reduce access block for post-operative care. In this study, the improvement intervention is revisited 5 years on with the objective to see if the changes are sustained and to understand factors contributing to sustainability.

**Design** A mixed-method approach was used, with quantitative analysis of ICU administrative data and qualitative analysis of interviews with hospital management and ICU staff.

**Setting** ICU with mixed surgical and non-surgical cases in a regional hospital in Australia.

**Participants** Interview participants: ICU nurses (four), ICU doctors (four) and hospital management (four).

**Main outcome measures** Monthly number of elective surgeries were cancelled due to unavailability of ICU beds. Staff perceptions of the interventions and factors contributed to sustainability.

**Results** After a decline in elective surgeries being cancelled in the first year after the intervention, there was an increase in cancellations in the following years ($\chi^2$=16.38, p=0.003). Lack of knowledge about the intervention and competitive interests in the management of patient flow were believed to be obstacles for sustained effects of the original intervention. So were communication deficiencies that were reported within the ICU and between ICU and other departments. There are discrepancies between how nurses and doctors use the escalation plan and regard the availability of ICU beds.

**Conclusion** Improvement interventions in healthcare that appear initially to be successful are not necessarily sustained over time, as was the case in this study. In healthcare, there is no such thing as a 'fix and forget' solution for interventions. Management commitment to support communication within and between microsystems, and to support healthcare staff understanding of the underlying reasons for intervention, are important implications for change and change management across healthcare systems.

## BACKGROUND

The long-term sustainability of interventions implemented in healthcare is important,

### Strength and limitations of this study

► This is a 5-year follow-up study of an implementation of an intervention.
► A mixed-method approach with both quantitative and qualitative analysis provides an important insight to both understand if the implementations had sustained or not and describe why it had or had not sustained.
► The mixed-method approach used made it possible to guide the interviews from the quantitative results, but also to verify some of what has been stated in interviews with the quantitative results.
► The statistical analysis suggests that there might be other factors that could affect the rate of cancelled surgeries such as staff and policies changes, or baseline increase in demand due to, for example, an ageing population.

but seldom evaluated. More typically, the effectiveness of an intervention is assessed shortly after implementation and then the evaluation is deemed to be complete.[1] Single interventions or one-off implementation of an intervention bundle may seem attractive, but they are largely unsuccessful in effecting meaningful change over time in clinical practice.[2] The definition of sustainability is somewhat controversial.[1 3] It can be described as something steady that does not revert, but also as a dynamic state where the intervention can adapt to new situations.[4 5] From a quality and patient safety viewpoint, the importance of an intervention should be the effect of it, not the intervention itself.

Bed occupancy at intensive care units (ICUs) can show unexpected large variations due to admission of critical patients. It is, therefore, a challenge to balance the capacity for planned elective surgeries requiring intensive care after surgery with the unplanned critical patients in need of intensive care,

particularly in hospitals with a single ICU for both categories of patients. In 2014, senior clinical staff at the ICU at a regional hospital in Australia saw a need to optimise the patient flow through the ICU to decrease cancellations of elective surgeries due to the unavailability of ICU beds.[6] An ICU escalation plan, a daily multidisciplinary morning handover meeting and optional education in system resilience for staff members were developed and implemented at the end of November 2014. The escalation plan, based on three different states—green, amber and red—was introduced as a system to indicate bed availability. After a follow-up period of 10 months, the intervention was found to have succeeded; since patient flow improved and elective surgeries cancelled due to unavailable ICU beds, the key performance indicator of the intervention was significantly reduced.[6] Improved internal communication, decision-making and cohesion within the ICU and better coordination between ICU and other hospital departments, such as surgery and the emergency department, were found to contribute to the success of the intervention.

Improvements in healthcare must be sustainable. Yet we know that healthcare practices often revert to previous practice once the energy and funding associated with the original intervention are removed.[1] Hence, it is critical to study the long-term sustainability of interventions in addition to the immediate outcomes. A review of evidence-based studies of intervention sustainability found that only 40%–60% of interventions are sustained over time.[7] Most of the studies focused on the sustainability of activities, only a few on performance indicator outcomes. In research on sustainability of interventions, it is important to study both the sustainment of the outcome of the intervention and of the practices.[1] It has been suggested that researcher site visits to interview multiple informants are valuable to study the sustainment of practices.[1]

The current study revisited the implementation of an ICU escalation plan and morning meeting at the same regional hospital 5 years on, with the objectives:
1. To find if the effect of the 2014 implementation has been sustained over time or not.
2. To understand why the effect of implementation has been sustained or not.

The setting is a 14-bed ICU with mixed surgical and non-surgical cases. There have been no changes in the number of beds, no significant changes in staffing and no other patient flow interventions, in the ICU during the studied period.

## METHOD
We conducted a mixed-method study. First, quantitative data were collected to examine the effect of the intervention over time. Second, qualitative data on staff perspectives regarding the intervention's sustainability was collected.

## Quantitative method
ICU audit data were collected from July 2014 to December 2019. The monthly number of cancelled surgeries due to no available ICU beds was collected to understand if the initial implementation had sustained or not. The monthly number of planned elective surgeries demanding ICU beds, nursing hours per patient day (HPPD; a measure of workload for nurses),[8] bed occupancy and length of stay (LoS) at the ICU was collected to decide if they influenced cancellation rate. Bed status (green, amber and red) was determined from notes recorded at the ICU morning meetings and from referrals, refusals and medical emergency calls to assess the compliance to bed status registration.

The relationship between the surgeries planned that required an ICU bed and the elective surgeries cancelled because of the ICU being at full capacity was examined using a $\chi^2$ test of independence, a p value of <0.05 was considered statistically significant.

A p-chart was calculated based on monthly numbers of cancellations to show the trend of cancellations. Since too many months had zero cancellations to allow for the analyses to be made by month, quarters (3 consecutive months) were used as the base for the analyses, starting from the first month after the intervention.

To determine if there were baseline changes in ICU demand or workload over the study period that could influence the result of the key performance indicator and cancellations of surgery, simple linear regression analyses were performed using data on HPPD, ICU bed occupancy, average LoS and monthly number planned elective surgeries. To account for the number of analyses in the linear regression analyses, an adjusted p value of <0.0125 was considered statistically significant.

Descriptive statistics (number and percentage) were used to tabulate results. SPSS (V.27) and SPC XL 2020 (V.2.50.0600) were used for statistical analysis.

## Qualitative method
Semi-structured interviews were held with ICU nurses, ICU doctors and hospital management 5 years post-implementation. The interviews were conducted by ER, as part of a research project for her studies at medical faculty after training by RC-W and EEA that are experienced in interviewing techniques. The staff members were approached, in person, by telephone or mail, and by one of the research team, and asked to participate. They were chosen by convenience, while ensuring that a mix of leaders and clinicians from the hospital and ICU were included. None of the participants knew the interviewer before the study, they were informed about the goal with the interviews and research project, and all consented to be interviewed. All but two participants had previous knowledge about the 2014 implementation. The number of interviews was based on the number of interviews needed for data saturation in the previous study.[6] The interview guide was informed by the result of the quantitative analysis and designed to collect the staff member's perception

**Table 1** Surgeries with postoperative ICU recovery, planned and cancelled

| | Planned elective surgeries requiring an ICU bed, n | Cancelled surgeries due unavailability of an ICU bed, n (%) |
|---|---|---|
| First year after intervention (December 2014–November 2015) | 681 | 8 (1.17) |
| Year 2 (December 2015–November 2016) | 580 | 18 (3.10) |
| Year 3 (December 2016–November 2017) | 560 | 27 (4.82) |
| Year 4 (December 2017–November 2018) | 736 | 27 (3.67) |
| Year 5 (December 2018–November 2019) | 742 | 33 (4.45) |

ICU, intensive care unit.

of the plan and problems surrounding the implementation of the intervention (online supplemental appendix 1). The interviews were held at the interviewees' workplace with only the interviewer present. They were digitally recorded and professionally transcribed verbatim. Inductive interpretive analysis was performed to find key themes of the transcripts.[9] One researcher (ER) made the first analysis in which codes, subthemes and themes were developed. Codes were checked with two other researchers (RC-W and EEA), who separately coded one of the interviews, and consensus was reached through discussion. A second researcher (AR) read all transcripts and verified the key themes from the analysis. Themes and subthemes were validated through feedback and discussion with a hospital executive and ICU consultant doctor (PL), who is familiar with the ICU and the intervention over its full implementation period.

**Patient involvement**

Patients or other members of the public were not involved in this study.

## RESULTS

### Quantitative results

The mean cancellation rate was 6.9% in the 5 months preceding the intervention. After a decline in elective surgeries being cancelled per year in the first year after the intervention, there was an increase in cancellations in the following years ($\chi^2$ (4, N=3299)=16.38, p=0.003). The percentage of cancelled surgeries per year increased, beginning from 1 year post-intervention, and has since maintained between 3.1% and 4.8%. The number of cancelled surgeries tripled in the years following the first year after the intervention (table 1). The trend of the

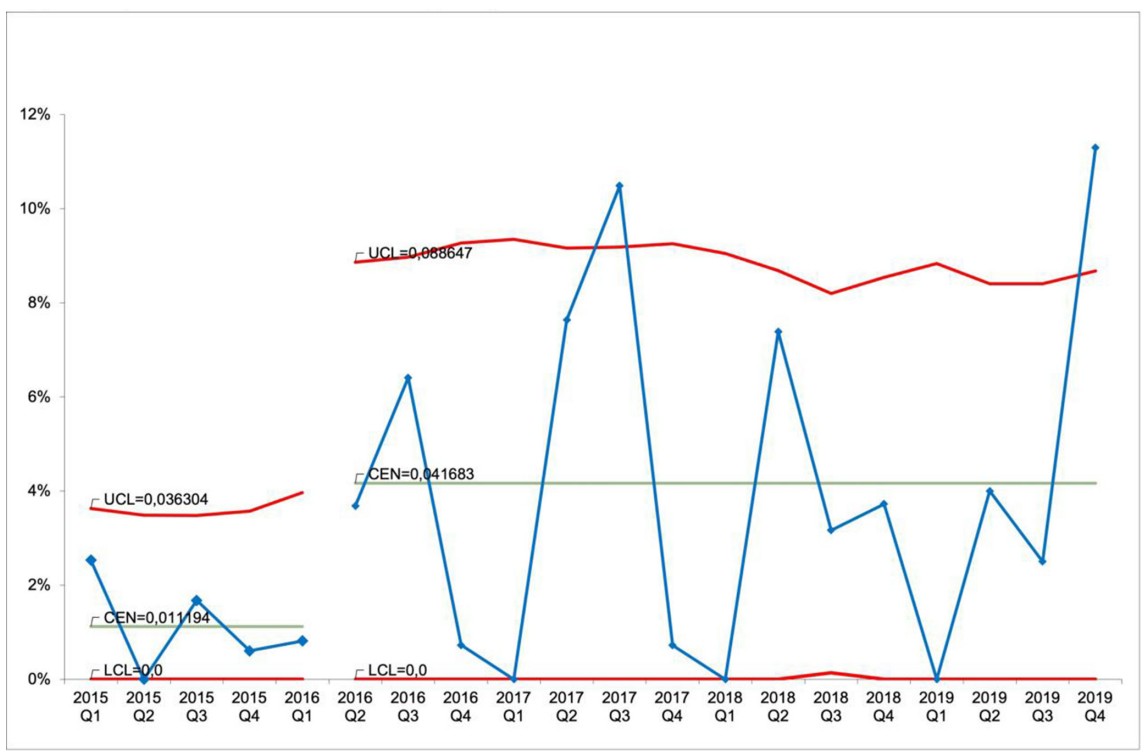

**Figure 1** Proportion of planned surgeries that were cancelled due to no available intensive care unit beds per quarter, p-chart. Q1 is the first quarter in the study period, December 2014–February 2015.

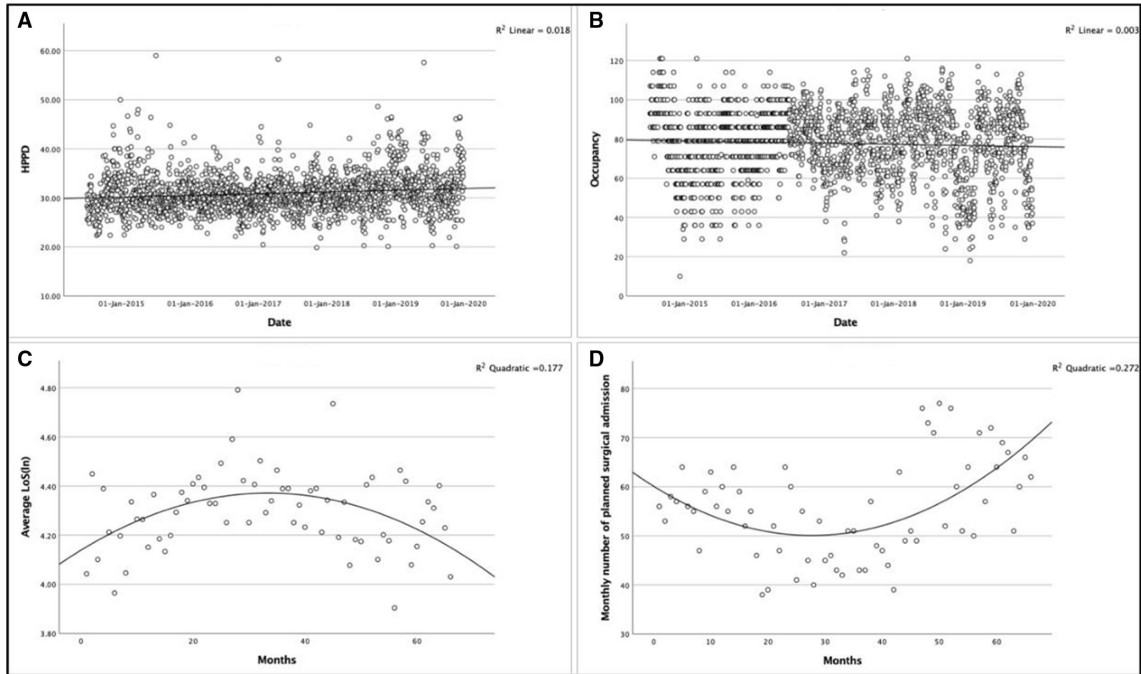

**Figure 2** Changes in intensive care unit (ICU) demand or workload over the study period. (A) Hours per patient day is a measure of workload for nurses, per date. (B) Occupancy at ICU, per date. (C) Average length of stay at ICU, per month. (D) Monthly number of planned surgical admissions to ICU, per month.

cancellation rate is shown in a control chart in figure 1. The high variability around the mean of cancellations in the period after the initial improvement implies that the process of cancellations of surgery is unstable.

In the analysis of baseline changes in ICU demand or workload over the study period, all variables were normally distributed except for average LoS, which was normally distributed after logarithmic transformation. HPPD has increased over the years since 2014, $(F(1,1982)=35.99, p<0.001)$, with an $R^2$ of 0.018 (figure 2A). For occupancy, the regression equation was not significant $(F(1,1982)=5.21, p=0.023)$ (figure 2B). The scatterplot of occupancy by date suggests that there is a discrepancy in how the data were recorded prior to mid-2016 as the data points are grouped around distinct values. There is, however, no clear pattern and should, therefore, not bias the results.

The data for both average LoS (figure 2C) and the number of planned surgeries requiring ICU beds (figure 2D) indicate that a curvilinear relationship fits the model better than a linear relationship. Therefore, the linear model was transformed by including a square of the independent variable 'month' in the linear regression equation. This polynomial regression produces two regression coefficients in addition to the intercept. For average LoS, the coefficient for the linear variable was positive and significant $(p=0.001)$, and the coefficient for the quadratic variable was negative and significant $(p<0.001)$. This indicates that average LoS initially increased until a turning point after which it decreased over time $(F(2,63)=6.76$, adjusted $R^2=0.15)$. For the number of planned surgeries requiring ICU beds, the

coefficient for the linear variable was negative and significant $(p=0.002)$, while the coefficient for the quadratic variable was positive $(p<0.001)$ suggesting that the relationship is negative until a turning point after which the relationship is positive $(F(2,63)=11.79$, adjusted $R^2=0.27)$.

Daily bed status registration was observed in notes recorded at the ICU morning meetings and from referrals, refusals and medical emergency calls. However, the data were incomplete from October 2016 to May 2018 and completely missing from July 2017 to February 2018. According to hospital officials, bed status was registered in that period; however, registration data are lost.

## Qualitative results

The multidisciplinary morning meetings implemented as part of the original intervention are still running at 08:00 every morning and the bed status is still recorded for the escalation plan in the meeting. Twelve face-to-face interviews were conducted, four in each group—ICU doctors, ICU nurses and hospital managers. Interview length ranged between 18 min and 46 min (average 30 min).

## Themes

Data saturation was achieved since no new themes emerged from the later interviews. In the analysis, two themes with associated subthemes emerged: sustained benefits of implementation, and factors adversely affecting sustainability (table 2). For the complete coding schedule, see online supplemental appendix 2.

**Table 2** Themes and subthemes found in the analysis in the interviews

| Themes | Subthemes |
|---|---|
| Sustained benefits of implementation | Positive experiences |
| | Enabling factors |
| Factors adversely affecting sustainability | Management of operations |
| | Lack of communication and understanding |
| | Plans not working as intended |
| | Reality of clinical work |

## Sustained benefits of implementation
### Positive experiences

In a busy environment such as an ICU, where many professions work together to solve problems, teamwork is vital. In the interviews, examples were given that the morning meeting supported teamwork: 'It would ensure that if there was things came up like the pharmacy, that the patient was going to be getting their medications at the right time because they were made available' (manager 1). This also made the ICU '… a nicer place to work' (nurse 1).

Furthermore, the morning meeting was beneficial because it facilitated communication within the ICU as 'it is easy – effective transmission of knowledge' (doctor 1). The meetings facilitated work by bringing attention to what was going to happen during the day. For example, '… the pharmacist can start ordering specialist drugs if you need them …' (manager 1).

One benefit of the escalation plan was that it created a plan for how to operate when work was getting hectic since '… it's made us more aware of forward planning and being proactive' (manager 1). One example was when patients were transferred to a neighbouring hospital in accordance with the plan; another is that it increased the ICU consultants' authority to say no to elective surgeries when the ICU was under pressure.

### Enabling factors

That the morning meetings are the first activity of the day and are held with all staff together, enabled its continuation. It was '… the only point in time you can have everybody to sit down' (manager 1). The morning meetings were beneficial from a human resources perspective because they ensured that 'people come to work on time' (doctor 3). The visually displayed feedback of the colour-coded bed status was thought to be beneficial. One participant reported that 'it's easier to do it (record bed status) because we are doing it in a handover meeting in the morning' (doctor 2). The implementation worked well for a while because 'everybody was on the right page' (nurse 1), and '… it worked because it was something everyone was willing to try' (doctor 4).

## Factors adversely affecting sustainability
### Management of operations

Most participants reported that the pressure for elective surgeries has increased over the last 5 years and that there is a need for more resources to be allocated for the ICU. For example, 'there's a massive push to meet our NEST (National Elective Surgery) targets and we are behind …' (doctor 1). And 'If there's no more money for ICU then we just have to accept that there's going to be more cancellations and longer delays, that's reality' (manager 3).

Some participants expressed that there were hierarchical barriers that created a negative workplace culture, '… there are good leaders and there are bad leaders and there are good leaders that help people progress … Then there are some people that just put barriers in front of people …' (nurse 2). All but two participants believed that the leadership do not understand the work done by frontline staff. One nurse expressed this as 'I think people are actually considering themselves a number' (nurse 2).

### Lack of communnication and understanding

A lack of transparency about how nurses were rostered onto shifts was expressed, 'there has been loss of transparency about the staffing that we do have in the unit … and that actual discussion about where we are, has been lost' (doctor 1). Furthermore, there was a discrepancy between how nurses and doctors view the number of available beds. The nurses thought that they had the capacity to fill up the ICU with elective cases, by increasing the nurse staff with flex time, whereas the doctors wanted to keep beds free in anticipation of emergency cases. The communication problems also extended outside the ICU resulting in communication problems with other wards and up to management level. For example, 'They have different booking staff down there that don't really know what goes on up here' (nurse 1), and '…ICU need to speak the same language with the executive and surgical service group managers. We are not speaking the same language' (doctor 2). It was also reported that some staff are unfamiliar with the escalation plan. Bed status was only visual at morning meetings and, therefore, 'there is no eyeballing of that' (doctor 1), which makes it easier to forget. On occasion, the staff did not understand the point of the escalation plan, 'So the numbers are actually useless. The colour code is useless. It's only for internal use.' (doctor 2).

### Plans not working as intended

Over the 5 years since it was introduced, the use of the escalation plan has changed. There are several examples of when the staff member needs to go the extra length to perform their duties, 'I can't count the number of times I have brought my food to the bedside to eat' (nurse 4). Some participants expressed that the escalation plan was not comprehensive enough and that it should include a plan for how to respond to emergency admissions.

Bed status was not filled out in the afternoon as regularly as was intended, 'because busy-ness, and the setting makes it easier to forget' (doctor 2). Several examples were given when the colour coding system did not work as intended, 'it's called green when we should be amber, it's called amber when we should be red' (doctor 3), and 'there was no major result from' escalating it up to management level (manager 2).

High workload predisposes for a workplace with a negative culture where 'Everyone's glum. There's no smiley. No happiness. No nothing and everything goes wrong.' (nurse 2), and '… we couldn't bring the patient to ICU because there was no nurse to look after the patient. … I actually took time off sick, stress leave …' (doctor 4).

### Reality of clinical work
Lastly, some factors emerged over the years after the intervention that the escalation plan either did not foresee or was unable to accommodate, for example, the introduction of electronic medical records and delayed ward discharges. A common problem at ICUs is large variations in workload, and they are '… either flooded or dry as we never get that rhythm happening for any period of time' (manager 2).

A common view was that the reason for the escalation plan not working was that there were not enough nurses and that 'they're not always available when you need them.' (doctor 4). A high staff turn-over rate with many newly recruited nurses was mentioned as detrimental for the work in the ICU 'we struggle, because we started to take graduates as well. They are only doubled up for a number of weeks and then we've got critically ill patients. That takes a lot more of the float nurse's time …' (nurse 4).

### DISCUSSION
This study is a 5-year follow-up study, which gives insight into the long-term effect after implementation of an ICU patient flow intervention. To investigate if changes have been sustained, the evaluation should be done at least 1 year after the intervention.[1 10] The intervention in 2014 in itself has continued to exist (the morning meeting, the bed status registration and the escalation plan), but the combined positive effects of the implementation have not been sustained over time. More elective surgeries are cancelled due to the unavailability of ICU beds from the second post-intervention year. The increase of cancelled elective surgeries is not explained by a higher bed occupancy or increase in the average LoS at the ICU. However, planned elective surgeries have increased and HPPD has significantly increased. This indicates that both demand for ICU post-surgical beds and the workload for the nurses has increased and that the increased pressure for elective surgeries that the participants expressed in the interviews was indeed true. The high variability of surgery cancellation rates a few years after the implementation is indicating a non-functioning process.

A major finding in the study is the lack of communication within ICU and between ICU and other departments, within the microsystem and between the microsystems and mesosystems. Healthcare can be described as a complex adaptive system (CAS) with microsystems, mesosystems and macrosystems.[11 12] Microsystems are the smaller units that are formed around the patient at the point of care.[13 14] The different microsystems are connected to and affect each other, through the mesosystem. In CAS, the communication between components affects system performance; for risk management, the microsystems and mesosystems have to continually coordinate their responses to adapt to changing conditions.[15] The reported deficiencies in communication within ICU and between ICU and other departments may hence be contributing to the positive effects of the implementation that were not sustained over time.

Workplace culture issues adversely affected sustainability in several ways. For example, discrepancies were found between how nurses and doctors saw the availability of ICU beds and in that there were hierarchical barriers, this combined can create a negative workplace environment. A 3-year follow-up study by Greenhalgh et al[16] found that good interrelations in the work environment are important for sustaining implementation. They stated that when there is a lack of transparency with decision-making and workways, the communication is lost, and the microsystems cannot cooperate in harmony, which accords with our findings.

The multidisciplinary meetings have more positive effects associated with them than the escalation plan. That the morning meetings are always at the same time and in the same room is habit-forming, which can be an important element in sustaining change.[17] Several aspects of the meeting, such as the opportunity to review patients and communicate plans, support daily work in the ICU. As was also noted in the original study, the morning meeting facilitates coordination of the workday for several different professionals at the same time and reinforces teamwork.[6] These positive effects of the meetings improve the conditions within the ICU and could explain why they have become an integrated part of daily work.[18] Even though negative for the cancellation rate, one of the positive effects of the escalation plan is that it increased the staffs' authority to cancel planned surgery, thus empowering the microsystem, which was also expressed in the original study.[6]

Even though most of the participants worked at the ICU during the implementation in 2014, there seems to have been a lack of understanding of the necessity with the escalation plan. The original study found that there was no consistent agreement on the purpose of the plan,[6] and this could explain why the commitment and dedication to the escalation plan has been lost over the years. If the benefits of the intervention are not readily perceived by the staff the chance of sustainability is lower.[7] For improvement leaders, management or engaged clinicians, it is important to be able to explain the purpose of the intervention, and to support

knowledge and understanding among staff for long-term good quality outcomes.[7 19] Furthermore, Staines *et al* noted in a study of the long-term success of quality work in a healthcare organisation the importance of leadership commitment to improvement work and support to frontline staff work with improvements.[20] Thus, continuous follow-up by management to reinforce commitment is needed for a change to be fully accepted and firmly established at the workplace.

Several participants thought that the escalation plan was not sufficiently comprehensive to cover all situations in the real clinical work, and that this were an obstacle to the implementation of the plan. However, when new processes are introduced in CAS, it cannot be expected that they will stay the same since the conditions are always changing. That an intervention can be modified over time is influencing the extent of sustainability.[7] It is, therefore, important to acknowledge the difference between work-as-imagined (WAI) and work-as-done (WAD).[21] It might be that the escalation plan, being WAI, is not flexible enough for the dynamics in the actual work going on in the ICU (ie, WAD). One example is situations reported when the escalation plan was not used to protocol since the bed status classification did not fit reality. The original study, however, highlighted how the plan was co-opted by staff members in different ways, according to their own needs, suggesting that the plan was sufficiently flexible when it was first implemented.[6] Some of that flexibility may have been lost when the understanding and knowledge about the rationale for the escalation plan was lost. Interventions must be able to adapt to the changing context of healthcare to continuously refine and improve interventions to ensure they can be sustained.[22]

## Study limitations

Besides bed status registration, and the report that daily morning meetings are ongoing, there are no hospital data for an audit of compliance to all parts of the intervention in retrospect. Furthermore, in the regression analysis, the $R^2$ is small for most analyses, which suggests that the independent variable is only explaining a part of the variance in the outcome variable. Thus, there may be other factors that also explain the variance in the number of cancelled surgeries, for example, staff and policies changes not reported, or baseline increase in demand due to the ageing population. These factors could not be evaluated within the scope of this study.

## CONCLUSION

In this study, even though practices of the interventions have been sustained over time, the positive effects have not. Thus, there is no such thing as a 'fix and forget' solution for the implementation of interventions in CASs as healthcare. This study highlights important implications for sustainability of interventions to improve healthcare. Management commitment to support communication within and between microsystems, and to support healthcare staff understanding of the underlying reasons for intervention, is necessary for success over time after the initial enthusiasm with a new

intervention. Further studies on how this is achieved are warranted.

**Author affiliations**
¹Department of Biomedical and Clinical Sciences, Linköping University, Linköping, Sweden
²Australian Institute of Health Innovation, Macquarie University, Sydney, New South Wales, Australia
³Jönköping Academy for Improvement of Health and Welfare, The School of Health and Welfare, Jönköping University, Jönköping, Sweden
⁴Department of Anaesthesiology and Intensive Care, Linköping University Hospital, Linköping, Sweden
⁵Townsville Hospital and Health Service, Townsville, Queensland, Australia
⁶College of Medicine and Dentistry, James Cook University, Townsville, Queensland, Australia

**Contributors** All authors contributed to the planning and design of the study. ER collected the study data, and made the first analysis and interpretation, which were then finalised together with AR, RC-W and EEA. ER and AR drafted the manuscript. EEA, RC-W, LDG, PL and AJ revised it critically for intellectual content and approved the final version of the paper.

**Funding** The authors have not declared a specific grant for this research from any funding agency in the public, commercial or not-for-profit sectors.

**Competing interests** PL and AJ were involved in development of the intervention. They were, however, not interviewed as part of the study and were not involved in data collection or analysis. The authors have no other competing interests to declare.

**Patient consent for publication** Not required.

**Ethics approval** The Human Research Ethics Committee at the studied hospital (HREC/2019/QTHS/59796).

**Provenance and peer review** Not commissioned; externally peer reviewed.

**Data availability statement** All data relevant to the study are included in the article or uploaded as supplementary information. No data are available except those uploaded as supplementary information. Study data cannot be obtained by a third party as a requirement of ethics approval.

**ORCID iDs**
Eva Ros http://orcid.org/0000-0003-3256-3249
Axel Ros http://orcid.org/0000-0001-6302-8068
Elizabeth E Austin http://orcid.org/0000-0002-8438-2362
Robyn Clay-Williams http://orcid.org/0000-0002-6107-7445

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
