## [Reviewer comments · BMJ Open]

ARTICLE DETAILS

TITLE (PROVISIONAL)	Sustainment of a patient flow intervention in an Intensive Care Unit in a regional hospital in Australia: a mixed-method, five-year follow-up study
AUTHORS	Ros, Eva; Ros, Axel; Austin, Elizabeth; De Geer, Lina; Lane, Paul; Johnson, Andrew; Clay-Williams, Robyn

VERSION 1 – REVIEW

REVIEWER	Jerng, Jih-Shuin National Taiwan University Hospital, Internal Medicine
REVIEW RETURNED	17-Feb-2021

GENERAL COMMENTS	This manuscript described the results of revisiting an improvement intervention completed five years earlier to assess its sustainability. The results showed that the intervention effect failed to sustain within one year after the completion of the implementations. Although the report does have merits, several concerns are summarized below and need to be addressed. #1. Page 4, Line 38. Abstract-Results. The results provided only cancellation rate change and qualitative study results. The rest of the main quantitative outcome measures described in the Abstract were not mentioned in the Results. #2. Page 5, Line 17. Strength and limitations of this study. The 3rd bullet point stated that the study approach "verify" the quantitative results using qualitative methods. However, the interview was not designed based on the quantitative results, i.e., not a sequential study process. #3. Page 5, Line 49. Background. I recommend that the authors provide their reasoning on the sustainability of the intervention in relation to the performance of relevant key indicators of health care quality (the effect of an intervention, as describe in the manuscript). #4. Page 6, Line 14. Background. The description of " a follow-up period of 10 months" is somewhat ambiguous and not consistent with the periods described in Table 1. It might be stated as how many months the intervention period and the period of sustainability follow-up were, respectively. Again, what was the key performance indicator (such as surgery cancellation rate) that was long-termly monitored before and after the intervention? #5. Page 6, Line 14. background. This paragraph describes a general concept and is less relevant to sustainability; therefore, it might be removed or moved to the Discussion section. #6. Page 6, Line 50. Background. Causes of failure to sustain the effect of intervention might be indicated from the literature, which is essential to this study's design.
--

	#7. Page 7, Line 7. Method. The authors need to explain why they conducted a multi-method study instead of a mixed-methods study. Specifically, the qualitative and quantitative methods described in the manuscript appeared to be different projects, but the first paragraph of the Methods section appeared to aim at the same purpose. The relationship, or the integration, of the two methods described in this manuscript, requires further explanation. #8. Page 7, Line 11. Method. The authors need to provide necessary information of the unit(s)/hospital(s) receiving the intervention, such as type (surgical or mixed), bed number, and staff numbers. #9. Page 7, Line 16. Method. In the Qualitative method, the authors collected administrative and patient flow data. However, this section lacked the assessment of the compliance to the key processes related to the intervention, either from audits by the researchers or from the historical audit data of the unit. The authors also need to ensure no other intervention concerning patient flow management at the included ICU(s), which might affect the compliance to the original intervention. #10. Page 8, Line 46. Results. What was the average pre-intervention surgery cancellation rate? #11. Page 8, Line 50. Results. The surgery cancellation rate (before and after the intervention) might be shown with a control chart. This chart might more clearly demonstrate the trend of cancellation rate and a failed sustainability. As the number of cancellations was very small relative to the number of planned surgery, a g-chart might be considered if the actual cancellation dates are available across the several years. #12. Page 13, Line 5. Discussion. The authors might need to explain why the intervention was effective in 2014 and then became ineffective in 2015 because of a dramatic deterioration of operation management, communication, and hospital management after the intervention period. If these factors had already existed in the intervention period, why was the intervention successful? Also, for other factors "the intervention had not considered," did the interviews imply that these unknown factors emerged after the intervention? #13. Page 15, Line 46. Conclusion. I am afraid I have to disagree with the conclusion that the interventions are partly sustained. If the authors chose only one measurement for the intervention effect, such as surgery cancellation rate, the intervention effect failed to sustain. The "partly sustained" phrase was not seen in the Abstract. Besides, the second sentence, "Thus, there is no such thing..... as healthcare." appeared not supported by the research data. Results from the quantitative methods reached only one conclusion that interventions are not sustained, but the Methods and Results described regression analyses. This could perplex the readers about why the study deployed these regressions without reaching any conclusion.
--	---

REVIEWER	Ozga, Dorota Rzeszow University, Institute of Health Sciences, College of Medical Sciences
REVIEW RETURNED	05-Mar-2021

GENERAL COMMENTS	Dear authors, Thank you for the opportunity to review the article and I do so with satisfaction. Comments
---

	A summary and conclusions should be added to constitute a "signpost" for the reader and encourage the reader to further research in this area around the world. Supplement with bioethical principles.
--	--

REVIEWER	Sharma, Shrawan King's College London
REVIEW RETURNED	29-Mar-2021

GENERAL COMMENTS	Title of the paper: A multi-method study on the sustainment of an implementation: An ICU patient flow intervention revisited. Reviewer: Thank you for the opportunity to review the paper. The paper reported various factors influencing sustainment of a flow intervention aimed at improving availability of surgical beds in an ICU. Methodology is well described for qualitative data collection and analysis. Qualitative findings were presented through thematic analysis whilst quantitative findings were analysed using various data analysis techniques. The manuscript provides an interesting insight into how few years after a change implementation practice can be different and some factors that invariably affect long term sustainability of an intervention. The research subject is timely and relevant to the latest changes within wider health system. The findings from the research will be useful for healthcare administrators, various professionals and researchers interested in this field. I found that findings were reported with supporting quotes/evidence; however, there are several issues within the reporting structure. In addition, there are some sentences that need further clarity for better understanding of both research process and findings. I have provided recommendations below to enhance the quality of the manuscript. A focused alternative title of the paper: A mixed-method five years follow up study on sustainment of a patient flow intervention in an ICU/intensive care unit Please note page numbers are of manuscript not of the entire pdf submission. Abstract: [ ] I suggest you add a short sentence under a heading with background followed by another heading with aims and objective to understand the overall purpose of the study. [ ] I suggest using term mixed method which is widely used rather than multi method. Although multiple tools have been used as part of data collection however they fall under qualitative and quantitative methodology. This will hopefully also make your research more sensitive to search terms. [ ] Please add number of each professional group participating in the interview in
--

	brackets.  [ ] Your outcome measures in my opinion is only number of ICU bed available, cancelled surgeries and factor affecting sustainability. Please explain otherwise how you are measuring other indicators as a direct link to your study objectives. [ ] Outcome section sentences are very long with three and in one sentence. This affects readability of your abstract. [ ] You state in the sentence that this has implication for change but have not spelled out what is that implication. [ ] I suggest using 5 year follow up rather than long term as a term could be individual interpretation. [ ] In the sentence where you state a multi method approach....made it possible..... Readers will be interested to know if this is a strength or limitation. I suggest replace this with provides an important insight rather than made it possible. [ ] Last sentence of your strength and limitation heading in the abstract section- please explain this- someone who is only reading the abstract need to understand the implication of this on the study findings rather than in statistical terms. Background: Please rephrase the sentence below.. Therefore, the implementation of an intervention should be considered to be sustainable when the effects of the intervention have been sustained. do you mean considered to be successful ...? Page 6 row 13: I suggest you highlight the extent of the issues and key gains of the original study here, and improvement here so readers can make a judgement on how and whether this was successful. Page 6 row 21 to 41: In my opinion this paragraph appears stand alone and some of this may be useful to interlink with what you found in your study in discussion rather than in background. Page 6 row 58 to 60: These are very specific so objectives will be a better term to describe rather than aims. Page 7 row 1: I suggest replacing multi method with mixed method throughout the study. Page 7 row 16: data analysis should come after data collection methodology is described. Under quantitative method: With cancellation of surgery being dependent on bed status if bed status data was incomplete- why was it still useful to include this in your analysis? Please explain. There is a significant gap of nearly 2 years' worth information and presents challenge to the credibility of your findings. I suggest you state this in abstract with justification as to why the findings are still useful.
--	---

	Page 8 under result section: You report that more planned elective surgeries requiring an ICU bed have been cancelled per year... per year is an average, readers will be interested to know how was the decline/cancellation trajectory in a study where sustainability is examined. Page 9 row 1: You have reported HPPD as an outcome... this is more of an influencing factor in my opinion for sustainability. Please state in your methodology why it was useful to calculate and report. Qualitative results Page 9 row 40 to 50: Grammatical error in the first sentence...should be are still running rather than are still run. Please check journal writing guidance but in my view 12 should be written as twelve. This entire section from row 40 to 50 should go under data collection rather than results. Page 10 Positive experiences: grammar issues in first paragraph. Row 38: you reported under positive experience a sentence thatanother is that it increased the ICU consultants' authority to say no to elective surgeries. Please explain how this is a positive experience in sustainability study. Page 11 top row: Why the implementation has not been sustained- I suggest rephrase as factors adversely affecting sustainability and then rephrase headings of each paragraph to reflect negative impact of these issues as well as highlighting work culture as an issue that has been reported on many occasions but does not flow as a theme in any paragraph. Discussion: Page 12 top 12 rows: This is already said in your background; at this point I recommend you discuss findings, current evidence and their relevance in your context. Page 12 3rd paragraph: The sentenceA major finding in the study is the lack of communication within ICU and between ICU and other departments, i.e. within the... do you mean for example by writing i.e.? Implications for practice and recommendations: Many interested readers would want to find out what they should do to make change process effective. Please include these in your paper to enhance overall quality and applications of the findings into practice. General Comments: [ ] Grammatical error on several sentences need proof read before resubmission. [ ] Abstract, limitation and strength as well as discussion need a focused rewriting using work count carefully. I have given some recommendations. [ ] Discussion needs quantitative data integration to reflect effects on sustainability.
--	---

	[ ] Some statements/paragraph appears standalone and not in an order for manuscript that need attention. I have given suggestions in my detailed review above.
--	---

VERSION 1 – AUTHOR RESPONSE

Reviewer 1	Comments to the Author: This manuscript described the results of revisiting an improvement intervention completed five years earlier to assess its sustainability. The results showed that the intervention effect failed to sustain within one year after the completion of the implementations. Although the report does have merits, several concerns are summarized below and need to be addressed.
1:1.	Page 4, Line 38. Abstract-Results. The results provided only cancellation rate change and qualitative study results. The rest of the main quantitative outcome measures described in the Abstract were not mentioned in the Results. Authors' response We do not fully understand the comment. In the Results section in the main text, all quantitative measures in the study are provided. In the Results section in the Abstract we have chosen to provide cancellation rate, the most important quantitative result related to the aim of the study. In the new version of the manuscript we have also clarified that cancellation rate is the main outcome measure of the intervention and that other quantitative data are background results. Manuscript changes in: Abstract, Main outcome measures, page 2. Background, second paragraph, page 4. Method, first paragraph, pages 6.
1:2	Page 5, Line 17. Strength and limitations of this study. The 3rd bullet point stated that the study approach "verify" the quantitative results using qualitative methods. However, the interview was not designed based on the quantitative results, i.e., not a sequential study process. Authors' response Actually the interview questions were informed by the quantitative results, which is also stated in the third bullet point. We have now clarified this in the Methods section. Hence, we believe that the third bullet point is now appropriate. Manuscript changes in: Method, first paragraph, page 5. Method, Qualitative method, page 7.
1:3	Page 5, Line 49. Background. I recommend that the authors provide their reasoning on the sustainability of the intervention in relation to the performance of relevant key indicators of health care quality (the effect of an intervention, as describe in the manuscript). Authors' response

This is a good point and really the rationale behind the study. In the Background section of this article we aim to “set the scene” for this study, that it is important to follow up the long-term effects of an improvement intervention, and to study what factors influence the long-term sustainability of the interventions. Based on the result of the relevant key indicator, surgery cancellation rate, and results from interviews we provide our reasoning on the sustainability of the intervention in the Discussion.

Manuscript changes in:

We have added some comments regarding this in the Discussion, fifth and sixth paragraphs, page 15.

1:4.

Page 6, Line 14. Background. The description of "a follow-up period of 10 months" is somewhat ambiguous and not consistent with the periods described in Table 1. It might be stated as how many months the intervention period and the period of sustainability follow-up were, respectively. Again, what was the key performance indicator (such as surgery cancellation rate) that was long-termly monitored before and after the intervention?

Authors' response

The intervention was implemented in the end of November 2014, and the follow-up period started in December the same year. We have re-labeled the time periods in Table 1 to clarify. In the initial article (ref. 6) about the intervention, the results was described after a follow-up period of ten months, which is what we describe in the Background. The key performance indicator was indeed surgery cancellation rate which is also described, even if the words key performance indicator was not used, we have now added that in this section. In Table 1 we, for the present study, compare surgery cancellation rates for five years after the intervention.

Manuscript changes in:

Background, second paragraph, page 4.

In Table 1 the time periods are re-labeled, starting with “First year after implementation (Dec 2014 – Nov 2015), up to Year 5 ((Dec 2018 – Nov 2019).

1:5.

Page 6, Line 14. background. This paragraph describes a general concept and is less relevant to sustainability; therefore, it might be removed or moved to the Discussion section.

Authors' response

We believe that both the concepts of complex adaptive systems and of micro-, meso-, and macrosystems are important for the understanding of sustainability, and there were also results in the study supporting this. Hence, we believe it is important to mention/describe these concepts in the article. But we agree that it is more relevant to reflect on these concepts in the Discussion, and have now done so.

Manuscript changes in:

The paragraph is deleted from the background and the concepts are discussed in the Discussion, second paragraph, page 13.

1:6.

Page 6, Line 50. Background. Causes of failure to sustain the effect of intervention might be indicated from the literature, which is essential to this study's design.

Authors' response

Yes, causes of failure to sustain the effect of an intervention is important and the rationale for this study, which we designed to be able to describe important factors with an analysis of interviews with key persons for the studied care process. This question and our answer is related to 1:3. We

have now developed the manuscript relating to literature both in the Background to motivate the study design, and also in the Discussion when reflecting over our results.

Manuscript changes in:

Background, third paragraph, page 5.

We have also added some comments regarding this in the Discussion, fifth and sixth paragraphs, page 15.

1:7.

Page 7, Line 7. Method. The authors need to explain why they conducted a multi-method study instead of a mixed-methods study. Specifically, the qualitative and quantitative methods described in the manuscript appeared to be different projects, but the first paragraph of the Methods section appeared to aim at the same purpose. The relationship, or the integration, of the two methods described in this manuscript, requires further explanation.

Authors' response

The study is based on one project aiming at investigating the long-term effect of an improvement intervention. As described in the last paragraph in the Background there were two aims/objectives within the study: 1) To find if the effect of the 2014 implementation has been sustained over time or not, and 2) To understand why the effect of implementation has been sustained or not. To study these objectives both a quantitative and a qualitative approach was needed. This is also stated in the first paragraph in Methods.

The literature on multi- and mixed methods is rich and there are different views about the definitions and nomenclature. Since the first study on this intervention (ref 6), performed with similar methods as the present study and published in BMJ Open, was described as a multimethod study, we choose the same nomenclature for describing the present study. But since the outcome of the quantitative part of the study influenced the interviews in the qualitative part (comment 1:2), we agree it might be fair to say that this present study was indeed a mixed-methods study. This is also in line with our responses to reviewer 3 (item 3:2). We have thus changed the nomenclature from multi-method to mixed method where relevant in the article.

1:8.

Page 7, Line 11. Method. The authors need to provide necessary information of the unit(s)/hospital(s) receiving the intervention, such as type (surgical or mixed), bed number, and staff numbers.

Authors' response

We have added a paragraph in Background with relevant data adjacent to the objectives of the study. When it comes to staff numbers, numbers of nurses have changed slightly over the study years, in 2015 there were 117 full time equivalent nurses, in 2019 there were 121. Number of physicians has not changed at all. We judge this as that there have been no significant changes in staffing, which we now report in the revised manuscript, but do not see it necessary to provide exact numbers.

Manuscript changes in:

Background, last paragraph, page 5.

1:9.

Page 7, Line 16. Method. In the Qualitative method, the authors collected administrative and patient flow data. However, this section lacked the assessment of the compliance to the key processes related to the intervention, either from audits by the researchers or from the historical audit data of the unit. The authors also need to ensure no other intervention concerning patient flow management at the included ICU(s), which might affect the compliance to the original intervention.

Authors' response

The bed status registration allows for assessment of the compliance of one of the key processes. We have now clarified this in the manuscript. One other component of the intervention was the morning handover meeting, that since the intervention have taken place every morning. This is something you cannot evaluate in audit data, it is evaluated in the interviews in the study. There are no other hospital audit data on compliance to the ICU escalation plan. It would also have been impossible to study compliance more extensively in retrospect. We have added a notion about this in the Study limitations.

There were no other interventions regarding patient flow at the studied ICU during the study period.

Manuscript changes in:

Method, Quantitative method, first paragraph, page 6.

Study limitations, page 16.

1:10.

Page 8, Line 46. Results. What was the average pre-intervention surgery cancellation rate?

Authors' response

For a base-line period July-November 2014 it was 6,9% (20/288). This is now added in Results.

Manuscript changes in:

Results, Quantitative results first paragraph, page 7.

1:11.

Page 8, Line 50. Results. The surgery cancellation rate (before and after the intervention) might be shown with a control chart. This chart might more clearly demonstrate the trend of cancellation rate and a failed sustainability. As the number of cancellations was very small relative to the number of planned surgery, a g-chart might be considered if the actual cancellation dates are available across the several years.

Authors' response

Yes, we understand that it might be good to provide a control chart to illustrate the trend of the cancellation rate and the failed sustainability. For the data analyses for this study, we have access to data on monthly numbers of cancellations, and not actual dates for cancellations. Hence a g-chart cannot be used. For a p-chart there are too many months with no (zero) cancellations to allow for a p-chart analyses based on monthly numbers of cancellations. Hence, we have calculated a p-chart based on monthly cancellation numbers summarized by quarters (three consecutive months) for the whole study period from the intervention. The p-chart supports the result from the Chi-2 analyses presented in the manuscript. This is now addedo the article as a new Figure 1, with an explanation in Methods and a comment in Results and Discussion.

Manuscript changes in:

Methods, Quantitative method, third paragraph, page 6.

Results, Quantitative results, first paragraph, page 8.

Discussion, first paragraph, page 13.

Figure 1

1:12.

Page 13, Line 5. Discussion. The authors might need to explain why the intervention was effective in 2014 and then became ineffective in 2015 because of a dramatic deterioration of

operation management, communication, and hospital management after the intervention period. If these factors had already existed in the intervention period, why was the intervention successful? Also, for other factors "the intervention had not considered," did the interviews imply that these unknown factors emerged after the intervention?

Authors' response

Based on the result of the analyses of the interviews we reflect on the long-term effectiveness of interventions in the Discussion and conclude that several factors are important as stated in the Discussion and Conclusion. We do not believe that there is a dramatic deterioration in these factors, the issue is the ability to maintain these factors over time after the initial enthusiasm of the intervention has faded away. We have added a few words about this in the last sentence of the Conclusion.

Regarding 'other factors "the intervention had not considered," – this is described as a result from the interviews under the subheading/sub-theme Reality of clinical work as factors that emerged over the years after the intervention.

Manuscript changes in:

Conclusion, page 16.

1:13.

Page 15, Line 46. Conclusion. I am afraid I have to disagree with the conclusion that the interventions are partly sustained. If the authors chose only one measurement for the intervention effect, such as surgery cancellation rate, the intervention effect failed to sustain. The "partly sustained" phrase was not seen in the Abstract. Besides, the second sentence, "Thus, there is no such thing..... as healthcare." appeared not supported by the research data. Results from the quantitative methods reached only one conclusion that interventions are not sustained, but the Methods and Results described regression analyses. This could perplex the readers about why the study deployed these regressions without reaching any conclusion.

Authors' response

As stated in the first paragraph of the Discussion we argue that the interventions have been partly sustained in the sense that the morning meetings and the escalation plan are still in use, but that the effect of the intervention have not been sustained over time, with more elective surgeries cancelled. Since the wording "partly sustained" might be confusing we have changed the wording in the first sentence of the Conclusion.

We also argue that based on the qualitative result several factors including management commitment over time and good communication at the workplace is important for the longevity of effects of interventions. Hence there is no such thing as 'fix and forget' solutions for implementation of interventions in healthcare, which we believe is supported by the qualitative data.

The regression analyses were done to determine if there were baseline changes in ICU demand or workload over the study period that could influence the result of the key performance indicator, cancellations of surgery. This is discussed in the 6th paragraph in the Discussion, and is now clarified in Methods.

Manuscript changes in:

Method, Quantitative method, fourth paragraph, page 6.

Conclusion, page 16.

Reviewer 2

Dear authors,

Thank you for the opportunity to review the article and I do so with satisfaction.

2:1

A summary and conclusions should be added to constitute a "signpost" for the reader and encourage the reader to further research in this area around the world. Supplement with bioethical principles.

Authors' response

We are sorry but we do not understand the comment about bioethical principles. The study was performed under ethical permission as described in Methods.

We agree it may be nice to conclude with a phrase like "Further studies in the field are warranted" and have added a sentence about this.

Manuscript changes in:

Conclusion, page 16.

Reviewer 3:

Thank you for the opportunity to review the paper. The paper reported various factors influencing sustainment of a flow intervention aimed at improving availability of surgical beds in an ICU. Methodology is well described for qualitative data collection and analysis. Qualitative findings were presented through thematic analysis whilst quantitative findings were analysed using various data analysis techniques. The manuscript provides an interesting insight into how few years after a change implementation practice can be different and some factors that invariably affect long term sustainability of an intervention. The research subject is timely and relevant to the latest changes within wider health system. The findings from the research will be useful for healthcare administrators, various professionals and researchers interested in this field. I found that findings were reported with supporting quotes/evidence; however, there are several issues within the reporting structure. In addition, there are some sentences that need further clarity for better understanding of both research process and findings. I have provided recommendations below to enhance the quality of the manuscript.

A focused alternative title of the paper: A mixed-method five years follow up study on sustainment of a patient flow intervention in an ICU/intensive care unit

Authors' response

Thank you! Inspired by your title suggestion we have now changed the title to Sustainment of a patient flow intervention in an Intensive Care Unit in a regional hospital in Australia: a mixed-method, five-year follow-up study

Abstract:

3:1

I suggest you add a short sentence under a heading with background followed by another heading with aims and objective to understand the overall purpose of the study.

Authors' response

Even if this may be a good suggestion it is not in accordance with BMJ Open publication guidelines. We have however changed the wording under Objective in the Abstract in line with your suggestion.

Manuscript changes in:

Abstract, Objective, page 2.

3:2

I suggest using term mixed method which is widely used rather than multi method. Although multiple tools have been used as part of data collection however they fall under qualitative and

quantitative methodology. This will hopefully also make your research more sensitive to search terms.

Authors' response

Thank you for this suggestion, we have commented a similar comment from reviewer one (item 1:7). In the literature on multi- and mixed methods we find different views about the definitions and nomenclature. Since the first study on this intervention (ref 6), performed with similar methods as the present study and published in BMJ Open, was described as a multimethod study, we choose the same nomenclature for describing the present study. But since the outcome of the quantitative part of the study influenced the interviews in the qualitative part (comment 1:2), we agree it might be fair to say that this present study was indeed a mixed-methods study. We have thus changed the nomenclature from multi-method to mixed method where relevant in the article.

3:3

Please add number of each professional group participating in the interview in brackets.

Authors' response

Done.

Manuscript changes in:

Abstract, Participants, page 2.

3:4

Your outcome measures in my opinion is only number of ICU bed available, cancelled surgeries and factor affecting sustainability. Please explain otherwise how you are measuring other indicators as a direct link to your study objectives.

Authors' response

Thank you, we agree, and have changed accordingly in this section in the Abstract.

Manuscript changes in:

Abstract, Main outcome measures, page 2.

3:5

Outcome section sentences are very long with three and in one sentence. This affects readability of your abstract.

Authors' response

We agree, and have adjusted this section accordingly.

Manuscript changes in:

Abstract, Results, page 2.

3:6

You state in the sentence that this has implication for change but have not spelled out what is that implication.

Authors' response

We agree this should have been addressed and have now elaborated on the issue in the Conclusion.

Manuscript changes in:

Abstract, Conclusion, page 3.

3:7

I suggest using 5 year follow up rather than long term as a term could be individual interpretation.

Authors' response
Good point, adjusted.

Manuscript changes in:
Strength and limitations, page 3.

3:8

In the sentence where you state a multi method approach...made it possible.... Readers will be interested to know if this is a strength or limitation. I suggest replace this with provides an important insight rather than made it possible.

Authors response
Good point, changed as you suggested, thank you.

Manuscript changes in:
Strength and limitations, page 3.

3:9

Last sentence of your strength and limitation heading in the abstract section- please explain this- someone who is only reading the abstract need to understand the implication of this on the study findings rather than in statistical terms.

Authors' response.
We understand that it might be a bit theoretical, and we have changed accordingly.

Manuscript changes in:
Strength and limitations, page 3.

Background

3:10

Please rephrase the sentence below..

Therefore, the implementation of an intervention should be considered to be sustainable when the effects of the intervention have been sustained. do you mean considered to be successful ...?

Authors response
Yes, you are correct, we actually meant "successful". But we now think that that sentence is a bit superfluous, and does not ad anything to the paragraph, why we delete it from the manuscript.

Manuscript changes in:
Background, first paragraph, last sentence deleted, page 4.

3:11

Page 6 row 13: I suggest you highlight the extent of the issues and key gains of the original study here, and improvement here so readers can make a judgement on how and whether this was successful.

Authors' response
We have now clarified that the key performance indicator, cancellation rate, improved in the original study. Regarding the key gains, those that in the previous study was suggested to support the improvement; they are described in this paragraph, we have changed the wording to clarify this.

Manuscript changes in:
Background, second paragraph, page 4.

3:12

Page 6 row 21 to 41: In my opinion this paragraph appears stand alone and some of this may be useful to interlink with what you found in your study in discussion rather than in background.

Authors' response

A similar comment was provided by the first reviewer (item 1:5). We see your point and are now discussing the concepts mentioned in this paragraph in the Discussion instead.

Manuscript changes in:

The paragraph is deleted from the background and the concepts are discussed in the Discussion, second paragraph, page 13.

3:13

Page 6 row 58 to 60: These are very specific so objectives will be a better term to describe rather than aims.

Authors response

Good point, changed as you suggested, thank you.

Manuscript changes in:

Background, fourth paragraph, page 5.

3:14

Page 7 row 1: I suggest replacing multi method with mixed method throughout the study.

Authors' response

We have commented this above in item 3:2, and have replaced as you suggest throughout the manuscript.

3:15

Page 7 row 16: data analysis should come after data collection methodology is described.

Authors' response

Good point, changed as you suggested, thank you.

Manuscript changes in:

Method, Quantitative methods, pages 5-6.

3:16

Under quantitative method: With cancellation of surgery being dependent on bed status if bed status data was incomplete- why was it still useful to include this in your analysis? Please explain. There is a significant gap of nearly 2 years' worth information and presents challenge to the credibility of your findings. I suggest you state this in abstract with justification as to why the findings are still useful.

Authors' response

This item is related to your item 3:4 about outcome measures. Bed status registration is one part of the intervention, hence data about bed status registration should really be seen as one assessment of the compliance to the processes related to the intervention. For two of the years in the study registered data are incomplete. According to hospital officials, bed status was registered in that period, but registration data are lost. We have clarified and changed the description of this in the manuscript.

We do not think that this is such a substantial part in the study that it merits place in the restricted space in the Abstract.

Manuscript changes in: Method, Quantitative methods, first paragraph, page 6. Results, Quantitative results, fourth paragraph, pages 8-9.
3:17 Page 8 under result section: You report that more planned elective surgeries requiring an ICU bed have been cancelled per year... per year is an average, readers will be interested to know how was the decline/cancellation trajectory in a study where sustainability is examined. Authors' response Thank you, we agree. We have now added a control chart in a new Figure 1 that supports the result of the Chi-2 analyses and illustrates the trajectory. We have added text about this in the Methods and Results section. Manuscript changes in: Methods, Quantitative method, third paragraph, page 6. Results, Quantitative results, first paragraph, page 8. Discussion, first paragraph, page 13. Figure 1.
3:18 Page 9 row 1: You have reported HPPD as an outcome... this is more of an influencing factor in my opinion for sustainability. Please state in your methodology why it was useful to calculate and report. Authors' response This remark is similar to item 3:4. Yes, it is more of an influencing factor. We have changed the wording accordingly. Manuscript changes in: Methods, Quantitative method, first paragraph, page 6.
3:19 Page 9 row 40 to 50: Grammatical error in the first sentence...should be are still running rather than are still run. Please check journal writing guidance but in my view 12 should be written as twelve. This entire section from row 40 to 50 should go under data collection rather than results. Authors' response Thank you, we have changed the grammatical issue and the figures issue. When it comes to if these rows with a description of the interviews should be in Methods or Results there are different traditions, and in BMJ Open we find examples of both solutions. We choose to leave it in Results, since we believe they are easier read and understood there.
3:20 Page 10 Positive experiences: grammar issues in first paragraph. Authors' response We suppose you are referring to grammar issues in the quote in this paragraph (actually there are grammar issues in other quotes as well). Since these are the actual quotes used to illustrate our analyses, and we believe they are easy to understand, we do not think it is correct to change them.
3:21 Row 38: you reported under positive experience a sentence thatanother is that it increased

the ICU consultants' authority to say no to elective surgeries. Please explain how this is a positive experience in sustainability study.

Authors' response

Since one of the components in the intervention was to be able to act to lessen the burden on ICU staff, if the bed status was alarming, this is regarded as a positive experience. We have added a few sentences about this in the Results and Discussion to clarify this.

Manuscript changes in:

Results, Qualitative results, Positive experiences, third paragraph, page 10.

Discussion, fourth paragraph, page 14.

3:22

Page 11 top row: Why the implementation has not been sustained- I suggest rephrase as factors adversely affecting sustainability and then rephrase headings of each paragraph to reflect negative impact of these issues as well as highlighting work culture as an issue that has been reported on many occasions but does not flow as a theme in any paragraph.

Authors' response

The headings and subheadings in this section of the results are the themes and subthemes in the qualitative analysis. Based on your comment and suggestion we have evaluated our analyses once again. You are right, work culture, or rather workplace culture, is an important issue. Your advice on rephrasing this heading is good, thank you, we have changed that. Even though work culture issues are mentioned sometimes in the interviews and quotes, our analyses is that they together do not constitute a theme or subtheme. We have however highlighted the issue of workplace culture at some places both in the Results and the Discussion.

Manuscript changes in:

Results, Qualitative results, first paragraph on page 12.

Discussion, third paragraph, page 14.

3:23

Page 12 top 12 rows: This is already said in your background; at this point I recommend you discuss findings, current evidence and their relevance in your context.

Authors response

In the first paragraph of the Discussion we summarize the most important results, so that we can then discuss them related to the literature. We agree that the first sentence is already said in the Background and have changed according to the suggestion.

Manuscript changes in:

Discussion, first paragraph, page 12-13.

3:24

Page 12 3rd paragraph: The sentenceA major finding in the study is the lack of communication within ICU and between ICU and other departments, i.e. within the... do you mean for example by writing i.e.?

Authors' response

We understand that this might have been unclear and have changed accordingly.

Manuscript changes in:

This paragraph is now in total rewritten.

Discussion, second paragraph, page 14.

3:25

Implications for practice and recommendations: Many interested readers would want to find out what they should do to make change process effective. Please include these in your paper to enhance overall quality and applications of the findings into practice.

Authors' response

We have changed the last sentences in Conclusions to emphasise that our findings should have important implications in the practice of interventions to improve healthcare, and that further studies are warranted.

Manuscript changes in:
Conclusion, page 16.

3:26

General Comments:

- a) Grammatical error on several sentences need proof read before resubmission.
- b) Abstract, limitation and strength as well as discussion need a focused rewriting using word count carefully. I have given some recommendations.
- c) Discussion needs quantitative data integration to reflect effects on sustainability.
- d) Some statements/paragraph appears standalone and not in an order for manuscript that need attention. I have given suggestions in my detailed review above.

Authors' responses

a – d) We have addressed and changed where we regard it appropriate according to the issues you have raised and commented on them above. The manuscript have been proof read by the authors that are native English speaking.

VERSION 2 – REVIEW

REVIEWER	Jerng, Jih-Shuin National Taiwan University Hospital, Internal Medicine
REVIEW RETURNED	03-Jun-2021
GENERAL COMMENTS	The authors have optimally responded to the comments and adequately revised the manuscript.
REVIEWER	Ozga, Dorota Rzeszow University, Institute of Health Sciences, College of Medical Sciences
REVIEW RETURNED	26-May-2021
GENERAL COMMENTS	Well done, congratulations.
REVIEWER	Sharma, Shrawan King's College London
REVIEW RETURNED	08-Jun-2021
GENERAL COMMENTS	Thank you for the opportunity to review the paper. My previous suggestions have been addressed and the overall quality of reporting has improved significantly. I would be happy to recommend the paper for publication with a minor comment below;

	Under heading Conclusion: Page 47 line 36- This study highlights important implications for interventions to improve healthcare. Do you mean implications for sustainability of intervention? If so then please add. I would like to thank authors for their time and commitment in sharing their work with wider healthcare community.
--	--